# Radon exposure and COVID-19 mortality in pre-vaccination period: What links might exist?

Jean François Coudert[1], Ekaterina Dadachova[2]*,
Gilles Maignant[3,4], Stephanie Jonathan [3]*

1 C&C IngeScience, Mouans-Sartoux, France, 2 College of Pharmacy and Nutrition, University of Saskatchewan, Saskatoon, Canada, 3 Université Côte d'Azur, RETINES (Risques, Épidémiologie, Territoires, Informations, Education et Santé), Nice, France, 4 Université Côte d'Azur, Centre National de la Recherche Scientifique (CNRS), GREDEG (Groupe de Recherche en Droit, Économie, Gestion), Nice, France

* ekaterina.dadachova@usask.ca (ED); stephanie.jonathan@univ-cotedazur.fr (SJ)

## Abstract

Radon, a naturally occurring radioactive gas known for its health risks, has recently gained attention for its potential protective role against COVID-19 mortality. This cross-sectional ecological study examined the relationship between indoor radon exposure and COVID-19 mortality rates across eight countries, including several European nations, the United States, and the State of Kerala, India, during the pre-vaccination period. The analyzed data on the subject were derived from recent scientific publications. The environmental aspect was represented by the variable "indoor radon concentration or probability of exceeding a radon concentration in indoor air," depending on data availability. Using national radon surveys and COVID-19 mortality statistics, statistical analyses, including Spearman's correlation and Kendall Tau, were conducted between March and December 2020. The findings revealed a consistent negative correlation between radon concentrations and COVID-19 mortality rates, indicating that higher radon concentrations were associated with lower mortality rates. Regions such as Finland and Sweden, where radon exposure was relatively high, experienced significantly lower mortality. With Sweden and Finland showing a mortality risk reduction factor of respectively 1,42 and 5,47 during the first wave compared to the UK where Radon levels are very low. Although the findings are not overwhelmingly strong, the data suggest that radon exposure may have a mitigating effect on COVID-19 mortality.

## Introduction

One of the first outbreaks of the COVID-19 pandemic that appeared on March 1, 2020, in the department of Morbihan in western France [1] did not result in high mortality numbers, in contrast to the outbreaks which happened simultaneously in eastern France. It might be possible that an environmental factor could help to explain

**Data availability statement:** The raw data underlying this study come from publicly accessible sources, including: Données hospitalières relatives à l'épidémie de COVID-19 en France: https://www.data.gouv.fr/fr/datasets/donnees-hospitalieres-relatives-a-lepidemie-de-covid-19-en-france-1/. IRSN – Campagne nationale de mesure du radon: https://www.irsn.fr/fr/connaissances/Environnement/expertises-radioactivite-naturelle/radon/Pages/4-Campagne-nationale-mesure-radon.aspx. ECDC: https://www.ecdc.europa.eu/. EUROPA (EU): https://publications.jrc.ec.europa.eu/repository/. The New York Times – COVID-19 Data: https://github.com/nytimes/covid-19-data. U.S. Census Bureau – Population Density Data: https://www.census.gov/data/tables/time-series/dec/density-data-text.html. INSEE – https://www.insee.fr/fr/statistiques/4265390?sommaire=4265511. JRC – Joint Research Centre: http://publications.jrc.ec.europa.eu/repository/handle/JRC76737. Government of Kerala – https://dashboard.kerala.gov.in/covid/index.php. The processed dataset generated from these sources and used for the analyses reported in this article has been deposited in Zenodo and is now openly available without restriction: https://doi.org/10.5281/zenodo.16918330.

**Funding:** The author(s) received no specific funding for this work.

**Competing interests:** The authors have declared that no competing interests exist.

these observations. $^{222}$Radon ($^{222}$Rn), referred to in this paper as radon, is a radioactive noble gas with a physical half-life of 3.8 days. Radon, emitted naturally by the descendants of the Uranium natural, is naturally present in the soil of many territories of the world, including Morbihan in France. It concentrates in basements and floors of homes and buildings and spreads through the various ventilation systems. It can also be emitted directly by the construction materials used or by the beach sands used in the cements [2,3]. The significant body of published data related to thermal spas shows the therapeutic benefits of controlled exposure to radon for treating inflammatory diseases such as rheumatoid arthritis [4,5]. Recent publications on COVID-19 [6,7] show that exposure to another type of ionizing radiation, i.e., X-rays, resulted in impressive curative effects in COVID-19 patients.

Based on these data, we hypothesized that exposure to radon through inhalation at home (indoor) could constitute an environmental factor contributing to resistance against COVID-19. To evaluate the hypothesis that indoor radon exposure may result in reduced mortality of COVID-19, we performed an exploratory ecological study, where we analyzed aggregated data at the population level and identified statistically significant correlations (Spearman, Kendall Tau) between two parameters: $^{222}$Radon ($^{222}$Rn) concentration per m³ in the surroundings of residential buildings and the mortality rate of the COVID-19-infected population during the pandemic's pre-vaccination period. It is important to note that our observations relate only to COVID-19, which is an acute medical condition, and are in no way contradicting the correlations between radon exposure and lung cancer, which evolves decades after exposure.

This study is divided into four parts to explore the relationship between indoor radon exposure and COVID-19 mortality rates comprehensively. The first part qualitatively compares mortality rates and their trends in different European countries where indoor radon exposure is prominent. The second part explores the epidemiological data from Kerala, India, known for having some of the highest levels of radon exposure worldwide (High Background Natural Radioactive Areas-HBNRAs) [8]. The third part compares the average radon exposure limits previously accepted by health authorities with short-term radon exposure levels, whether or not they appear for therapeutic purposes in the world's numerous thermal baths, some of which have existed since ancient times. The final part is devoted to a statistical correlational study of COVID-19 mortality data from French departments and continental states in the USA. In conclusion, indoor radon exposure affects COVID-19 mortality and appears to be statistically valid based on the analysis conducted.

### Current definition of radon exposure

Exposure is defined as the concentration of radioactivity in the air we breathe. It is measured in Bequerels per cubic meter (Bq/m$^3$ in the SI system) or in PicoCurie per liter (pCi/L) in the traditional system (1 PicoCurie/L = 37 Bq/m$^3$). One Bq corresponds to one disintegration (radiation emission) of the radio-element per second. Radon radiation consists of alpha particles, which are $^4$He nuclei with 2 protons and 2 neutrons. Throughout this article, we will refer to indoor radon and its progeny, present in homes, as radon.

Radon, which is recognized as a carcinogen by the United Nations Scientific Committee on the Effects of Atomic Radiation (UNSCEAR) [9], is widely measured in indoor air, and most countries have defined exposure limit values above which remedial measures should be taken. The World Health Organization (WHO), the European Commission, and the Institut de Radioprotection et de Sûreté Nucléaire (IRSN) in France recommend a limit value of 300 Bq/m$^3$ averaged over one year of exposure [10]. In the USA, it is considered that residential buildings require remediation if values exceed 4 pCi/L (148 Bq/m$^3$) [11].

### Indoor radon measurements

The natural disintegration of uranium often leads to the accumulation of radon on the ground floors of dwellings through natural or mechanical ventilation systems (MVS). Geological, geophysical, and climatological variables interact in complex ways to influence this process. This complexity is described in the USGS/EPA Radon Potential Assessment Open File Report 93-292 [12], the European Radon Mapping Project [13], which compiles data from various European countries, and the French Atlas Radon [14]. Fig 1 [13] illustrates the complexity of these phenomena.

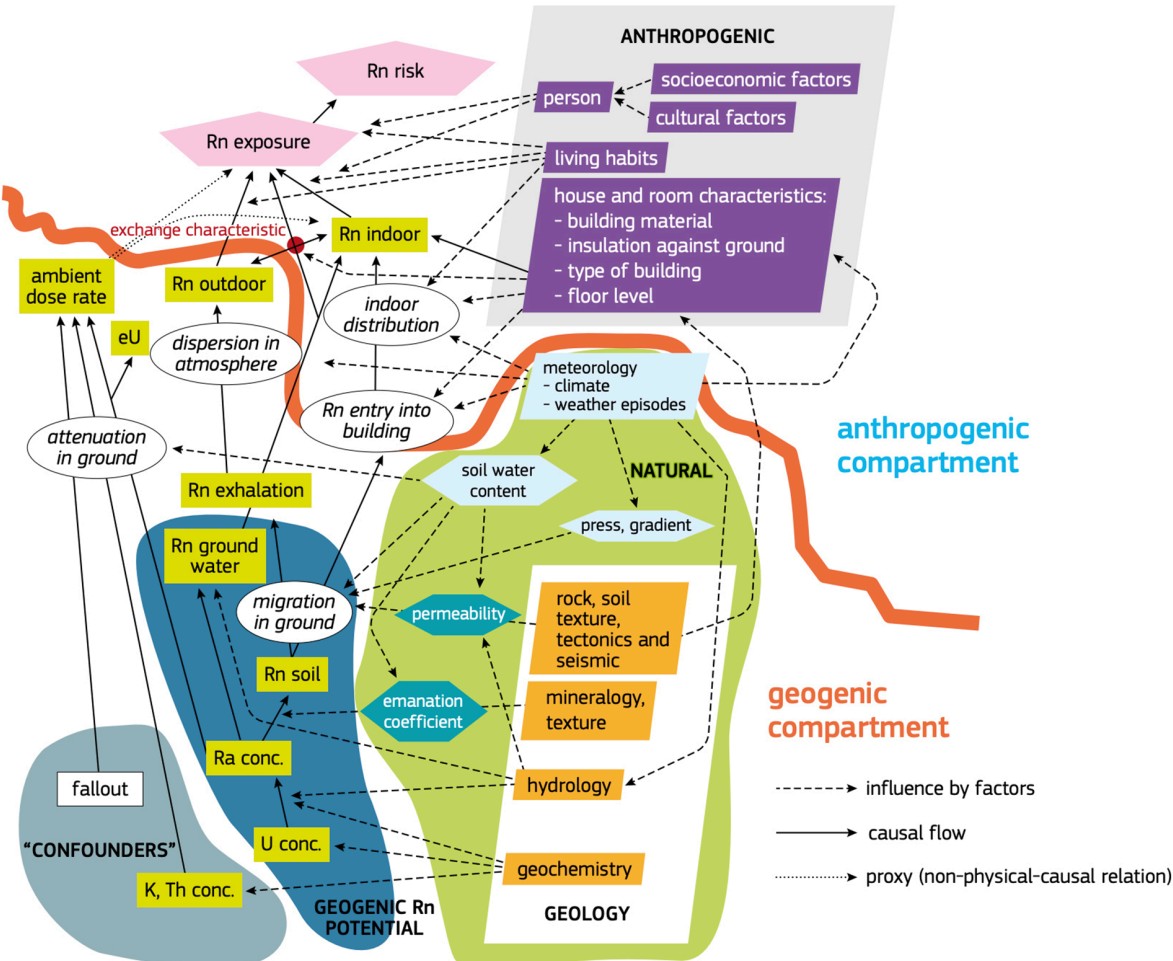

**Fig 1. Description of the phenomenon involved in the risk evaluation of the indoor radon.** Adapted from Bossew P. The European radon mapping project. Radiat Prot Dosimetry. 2013;157(3):522–530.

There are more than 100,000 home measurements available in the USA, 818,704 in Europe, and 12,641 in France. The indoor radon concentrations from these sources result from various processes, including measurement density per geographical entity, measurement methods, data compilation, and processing. This issue is discussed in detail in the European Project for the different countries [15]. For example, measurements in the USA use charcoal canisters in the lowest living area for two to seven days, while in France, measurements are taken over two months, mostly during the cold season, to account for climatological factors. The detailed procedures, data, and sources for each country's radon map (Fig 2) are provided in a European Commission-JRC publication [16]. The number of dwellings monitored up to 2005 is listed in Table 1.

All of these measurement campaigns have mobilized many experts in different countries over several years. They have been the subject of numerous international scientific exchanges and debates. These measurements are used as criteria for enforcing regulations and laws in various state governments. To our knowledge, they represent the best available data for this research. An individual's radon exposure also depends on sources beyond the local geology of their dwelling. These sources include exhalation from building materials that originate from other regions or countries [17].

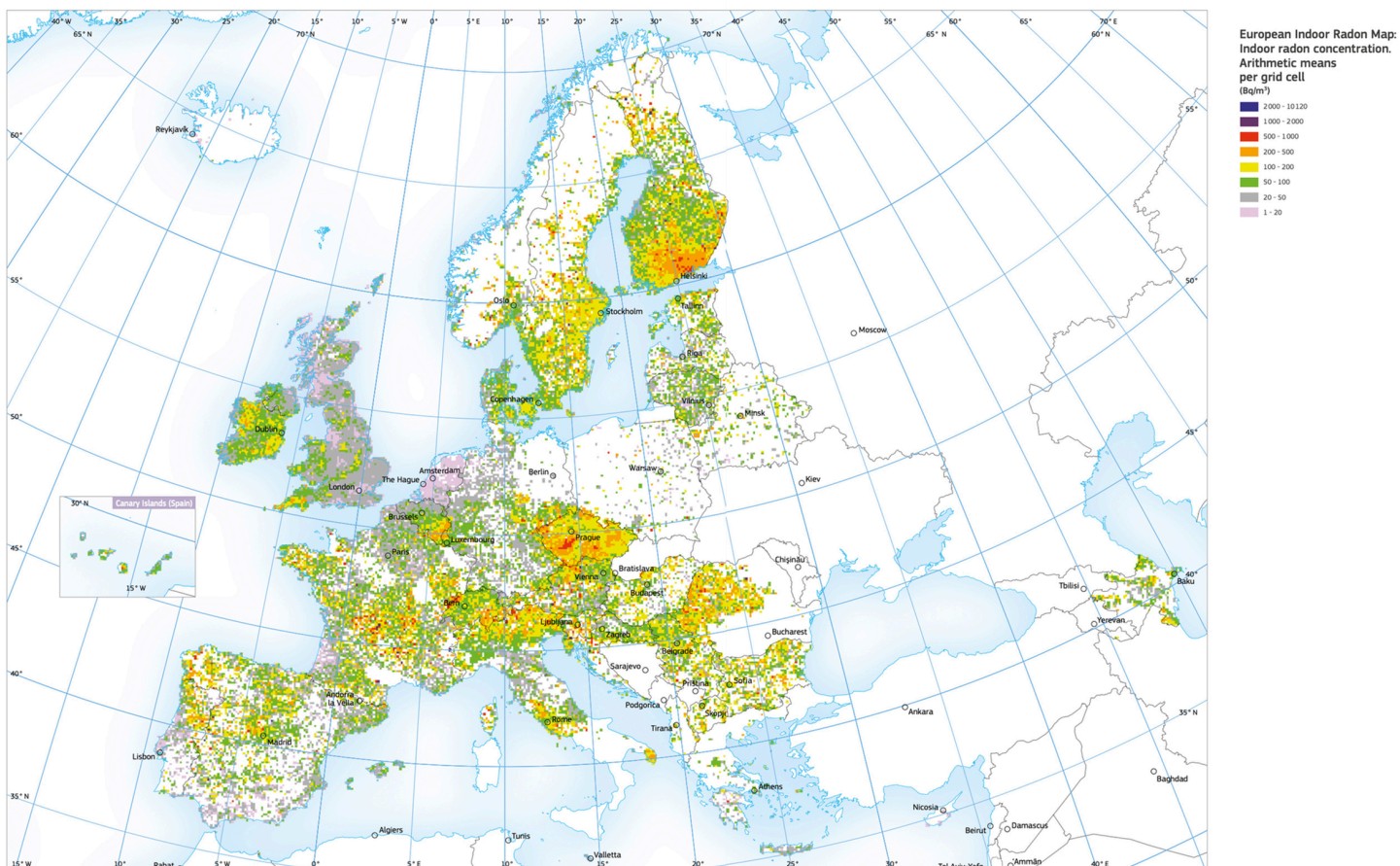

**Fig 2**. **Map of indoor radon in Europe, September 2019.** Adapted from European Commission, Joint Research Centre (JRC), Directorate G -Nuclear Safety & Security, REM project. Arithmetic means over 10 x 10 km cells of long-term radon concentration in ground-floor rooms.

**Table 1**. Number of dwellings monitored up to 2005 for each countries of the radon map.

| Country | Number of dwellings monitored | Country | Number of dwellings monitored |
|---|---|---|---|
| Albania | 110 | Latvia | 300 |
| Austria | 16000 | Lithuania | 400 |
| Belgium | 9000 | Luxembourg | 2619 |
| Croatia | 782 | Malta | 90 |
| Cyprus | 84 | Netherlands | 1846 |
| Czech Republic | 150000 | Norway | 51925 |
| Denmark | 3120 | Poland | 4098 |
| Estonia | 515 | Portugal | 3317 |
| Finlande | 73074 | Romania | 567 |
| France | 12261 | Serbia-Montenegro* | 968 |
| FYROM | NA | Slovakia | 4019 |
| Germany | >50000 | Slovenia | 2512 |
| Greece | 1277 | Spain | 5600 |
| Hungary | 15602 | Sweden | 500000 |
| Ireland | 11319 | Switzerland | 55000 |
| Italy | 5361 | | |

∗ Province of Vojvodina only.

## Comparison of mortality rates and exposure to indoor radon

To establish this comparison, we analyzed trends of the average deaths per 100,000 inhabitants—referred to as the "mortality rate"—from the beginning of the pandemic (mid-January 2020 to mid-December 2020) in countries with particularly high radon concentrations to derive a relevant indicator. The number of infections and recoveries cannot be used due to their dependence on the number of tests conducted. The study period concludes with the start of the vaccination campaigns.

The European Commission's Joint Research Centre (JRC) publishes the European Atlas of Natural Radiation, which maps indoor radon measurements in all European Union states [18]. This map is continuously updated and is created using a 10 km x 10 km grid (Fig 2), allowing for the identification of territories with high radon concentrations. To utilize this information, we focus on territories where mortality data related to the COVID-19 pandemic is available. This excludes areas at the intersections of several countries or political regions, such as the eastern part of France-Switzerland-Italy-Northern Austria, or areas that are too small, where COVID-19 data is not specifically geolocated or published. The United Kingdom, Sweden, and Finland were selected for this analysis.

The mortality curves for these three countries, calculated using data from the European Center for Disease Prevention and Control (ECDC), show significantly different mortality rates for the period from January 1, 2020 (week 1) to February 22, 2020 (week 59) (Fig 3).

This figure shows that throughout this period, Finland and Sweden exhibit higher radon exposure and lower mortality rates compared to the United Kingdom, which has minimal radon exposure (except in Scotland, Wales, and Cornwall).

To complete the comparison between these three countries, we collected the maximum mortality rates during the first and second waves of the pandemic, as shown in Table 2. Radon levels, measured as the Activity Median concentration in the air and expressed in becquerels per cubic meter (AM_Bq/m$^3$), are also reported for each country to provide context for environmental differences. We calculated the mortality rate ratio using the United Kingdom (0-50 AM_Bq/m$^3$) as the reference country. We compared this data with Sweden (50-200 AM_Bq/m$^3$) and Finland (50-1000 AM_Bq/m$^3$), both of which have significantly higher radon levels.

The analysis reveals significant differences in mortality rates correlated with varying levels of radon exposure. Specifically, in Sweden, the presence of radon without barrier or containment measures leads to a notable decreased in mortality risk by 30%, with a mortality pick value ratio of 0.70 (6.64/9.46) observed during the first wave of the pandemic.

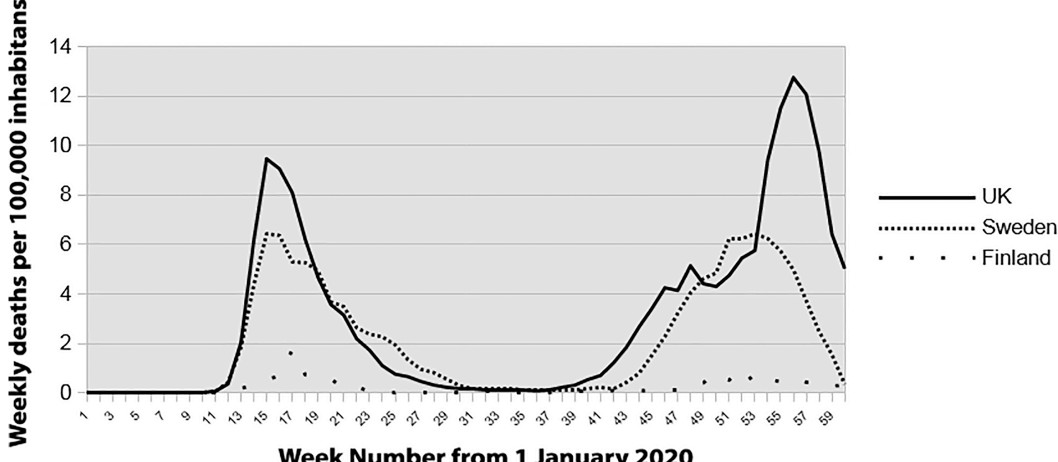

**Fig 3**. **Comparison of mortality rates for the United Kingdom, Sweden and Finland.** Source: European Centre for Disease Prevention and Control (ECDC). Available at: https://www.ecdc.europa.eu/en/publications-data/download-todays-data-geographic-distribution-covid-19-cases-worldwide.

**Table 2**. **Comparison of the Peak Values (PV ie maximum) of the weekly deaths per 100,000 inhabitants between United Kingdom, Sweden and Finland during the 1st and the 2nd wave of the pandemics.**

| Country | Peak Value | 1st Wave | *MR | 2nd Wave | MR |
|---|---|---|---|---|---|
| **United Kingdom** | Year-week n° | 2020-15 | | 2021-03 | |
| | Value | 9.46 | - | 12.75 | - |
| **Sweden** | Year-week n° | 2020-15 | | 2022-53 | |
| | Value | 6.64 | 0.70 | 6.43 | 0.50 |
| **Finland** | Year-week n° | 2020-17 | | 2021-01 | |
| | Value | 1.73 | 0.18 | 0.58 | 0.05 |

*Mortality Pick-value Ratio.
Source: ECDC, 2020.

This impact becomes even more pronounced when barrier and containment measures are implemented, as seen in Finland, where the mortality pick value ratio reduced substantially to 82%, with a mortality pick value ratio 0.18 (1.73/9.46).

Overall, the analysis of mortality rates in other countries with elevated radon exposure demonstrates consistent findings, indicating a clear suggestion of association between increased radon levels and lower mortality rates during the pandemic. This suggests that addressing radon exposure may be critical in mitigating health risks associated with COVID-19.

Several countries experienced virtually no first wave of mortality. To illustrate this point, we can analyze the case of the Czech Republic (Fig 4). Radon concentration is high and well distributed within its borders, as shown on the JRC map. The first COVID-19 case appeared on March 1, 2020, as in other European countries. However, the Czech Republic was not affected by the first wave, similar to other high radon-exposed countries like Sweden, Finland, and Kerala. Public policy regarding population protection in Czechia has been inconsistent, with a population not motivated by protective measures (e.g., barrier gestures, masks). This resulted in a first wave that occurred later when other European countries were already hit by the second wave.

The magnitude of the waves in Czechia was likely driven by spikes in infections due to this context. The result is striking, as shown in Fig 4. There was nearly no mortality during the first wave, with a rate of 0.66 for week 15, comparable to Finland's rate of 1.73. Other countries with high radon exposure, as shown on the Joint Research Center (JRC) map, also did not experience a significant first wave. These include Slovakia (Fig 4), with a mortality rate of 0.18 for week 16, and

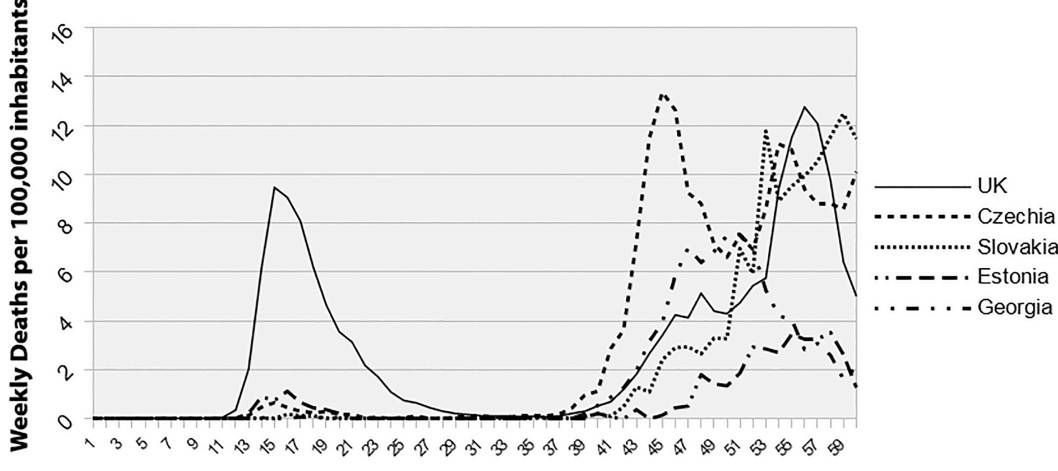

**Fig 4**. **Mortality rate for the UK, Czech Republic, Slovakia, Estonia, and Georgia.** Source: European Centre for Disease Prevention and Control (ECDC). Available at: https://www.ecdc.europa.eu/en/publications-data/download-todays-data-geographic-distribution-covid-19-cases-worldwide.

Estonia (Fig 4), where radon is prevalent on the most populated coast (Tallinn region), with a mortality rate of 1.3 in week 16. These countries exhibit similar mortality curves.

Outside Europe, data are available for Georgia (Fig 4), although the radon census there is incomplete, and it does not appear on the JRC map. Local studies [19] show areas of high radon concentration, particularly in the capital, Tbilisi, where the mortality rate remained very low.

In the next section, we discuss two aspects. The first relates to the statistical processing of French data to support the previous results, the second to more general considerations on the impact of radon.

## Materials and methods

### Statistical study using France data

To quantify the relationship between radon exposure (Fig 5) and the mortality rate, a statistical study was conducted in France (Table 3) on the correlation between this rate in each metropolitan department (90 departments) (source: Institut National de la Statistique et des Études Économiques - INSEE, 2020) and the average radon concentrations in these departments [14].

We find in Table 4 a quasi-constant correlation coefficient (–0.27) for random dates between 11/06/20 and 30/10/20, indicating that there are fewer deaths in areas with higher radon exposure (negative correlation coefficients).

### Statistical study from the United States of America

The mortality rates were obtained from data available on the Johns Hopkins University website [21]. In the early 1990s, the U.S. Geological Survey conducted a comprehensive radon survey for all states [12]. This study was based on detailed geological mapping, campaigns measuring uranium in the soil, and thousands of measurements taken from homes selected according to specific criteria, such as the type of dwellings, occupation types, statistical analysis, and the quality and precision of the measurements [22]. We, therefore, used the results of this study to perform correlation calculations. An example of the measurements conducted in the state of New Mexico based on each county is shown in Table 5.

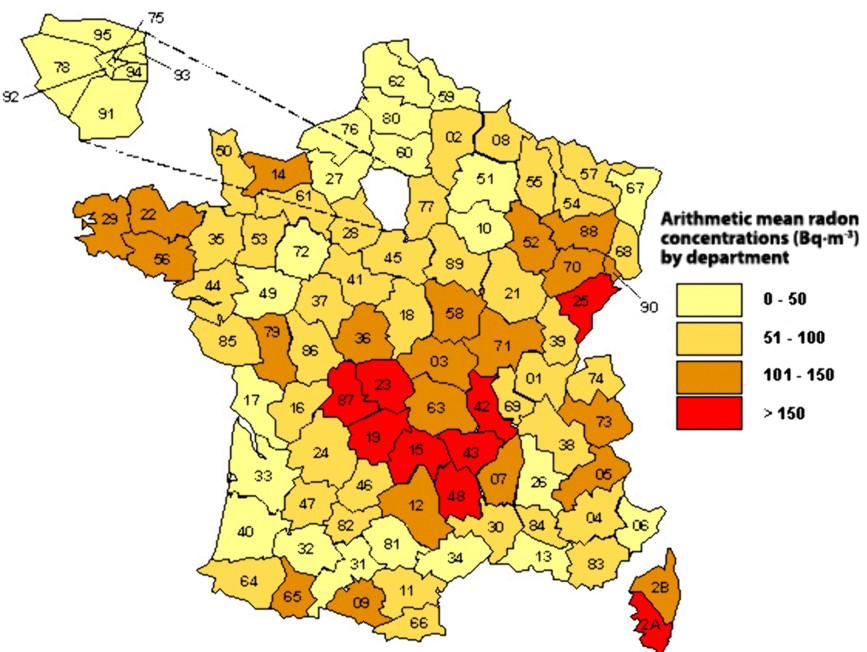

**Fig 5**. **Map of the average concentration of indoor radon measured by department.** Source: Institut de Radioprotection et de Sûreté Nucléaire (IRSN), 2000.

However, the methodology for compiling data at the state level is different from that used for French departments. We have a zoning map at this level (State/EPA Residential Radon Survey Screening Measurements), which shows the estimated percentage of houses with measured radon levels above 4 pCi/L (Fig 6). The results are divided into five classes, numbered from 1 to 5, representing percentages between 0-5%, 5-10%, 10-15%, 15-20%, and more than 20%. This is depicted on a map of the continental US plus Alaska, showing the percentage of homes with a screening level above 4 pCi/L.States are classified into 5 categories according to the "Estimated Percentage of Dwellings with Radon Levels Detected Above 4 pCi/L". States are assigned into 5 percentage classes [0-5], [5-10], [10-15], [15-20], [>20] pCi/L (Fig 6).

Instead of having a mean value for each state, as for the French departments, we have percentage classes for each state. To compare these results with those obtained for France, we calculated the Spearman correlation for four randomly selected dates (Tables 6 and 7). The states of New York and New Jersey were excluded from this analysis. Both states have the highest mortality rates of all states. New York State has 62 counties, and a detailed analysis shows that New York City has 5 counties and Greater New York has 8 more. Of these 13 counties, the most populous—comprising 12 million of the state's 22 million people— are all in class 1 (the lowest % radon exposure), while the entire state is in class 5 (the highest). The situation is similar in New Jersey, particularly in Jersey City, where 13 out of 21 counties are part of Greater New York with similar heterogeneity. A detailed analysis of these two states remains to be done.

The variable "radon" is presented as the percentage of exceedance of a reference value (in this case, 4 pCi/L), it represents a categorical ordinal variable which does not necessarily follow a normal distribution, so we turned to a Kendall rank correlation. The principle is the following: a first series is sorted and the ranks of the values of the second series are put in comparison with the first. For each observation, we note the number of subsequent values (noted B in Table 6) that are higher (+1) and lower (−1) than it (noted C in Table 6); this gives a new series of numbers that are balances between positive and negative numbers (S=C-B). All possible ordering of the [radon-class] pairs are considered i.e. binomial coefficient

**Table 3. Mortality rates and radon concentrations by France Department during the period of March to December, 2020.**

| Departments (Code) | Mortality rate | Radon conc. (Bq/m³) | Departments (Code) | Mortality rate | Radon conc. (Bq/m³) |
|---|---|---|---|---|---|
| Ain (01) | 16.5 | 55 | Lozère (48) | 1.3 | 264 |
| Aisne (02) | 53.9 | 62 | Maine-et-Loire (49) | 17.6 | 50 |
| Allier (03) | 12.1 | 145 | Manche (50) | 10.1 | 74 |
| Alpes-de-Haute-Provence (04) | 7.9 | 52 | Marne (51) | 48.2 | 41 |
| Hautes-Alpes (05) | 14.2 | 144 | Haute-Marne (52) | 51.2 | 136 |
| Alpes-Maritimes (06) | 17.8 | 37 | Mayenne (53) | 15.6 | 96 |
| Ardèche (07) | 35.6 | 134 | Meurthe-et-Moselle (54) | 49.9 | 61 |
| Ardennes (08) | 21.9 | 95 | Meuse (55) | 55.0 | 62 |
| Ariège (09) | 1.3 | 129 | Morbihan (56) | 12.8 | 142 |
| Aube (10) | 44.2 | 35 | Moselle (57) | 81.8 | 51 |
| Aude (11) | 15.9 | 86 | Nièvre (58) | 13.5 | 115 |
| Aveyron (12) | 8.6 | 101 | Nord (59) | 26.1 | 36 |
| Bouches-du-Rhône (13) | 28.9 | 32 | Oise (60) | 51.2 | 44 |
| Calvados (14) | 12.1 | 118 | Orne (61) | 15.5 | 94 |
| Cantal (15) | 6.2 | 161 | Pas-de-Calais (62) | 22.0 | 40 |
| Charente (16) | 3.7 | 90 | Puy-de-Dôme (63) | 6.9 | 146 |
| Charente-Maritime (17) | 7.9 | 45 | Pyrénées-Atlantiques (64) | 4.0 | 57 |
| Cher (18) | 27.9 | 79 | Hautes-Pyrénées (65) | 11.8 | 108 |
| Corrèze (19) | 15.3 | 217 | Pyrénées-Orientales (66) | 7.6 | 72 |
| Côte-d'or (21) | 47.0 | 76 | Bas-Rhin (67) | 61.0 | 38 |
| Côtes d'Armor (22) | 6.7 | 110 | Haut-Rhin (68) | 108.8 | 58 |
| Creuse (23) | 12.6 | 262 | Rhône (69) | 36.3 | 99 |
| Dordogne (24) | 3.4 | 79 | Haute-Saône (70) | 32.5 | 101 |
| Doubs (25) | 28.2 | 178 | Saône-et-Loire (71) | 36.5 | 116 |
| Drôme (26) | 25.0 | 36 | Sarthe (72) | 15.5 | 44 |
| Eure (27) | 14.1 | 45 | Savoie (73) | 17.4 | 114 |
| Eure-et-Loir (28) | 33.5 | 57 | Haute-Savoie (74) | 21.2 | 56 |
| Finistère (29) | 4.8 | 144 | Paris (75) | 82.7 | 22 |
| Corse-du-Sud (2A) | 30.5 | 263 | Seine-Maritime (76) | 14.7 | 45 |
| Haute-Corse (2B) | 6.2 | 133 | Seine-et-Marne (77) | 51.0 | 52 |
| Gard (30) | 13.3 | 84 | Yvelines (78) | 37.9 | 30 |
| Haute-Garonne (31) | 6.1 | 32 | Deux-Sèvres (79) | 5.9 | 103 |
| Gers (32) | 12.0 | 66 | Somme (80) | 40.7 | 42 |
| Gironde (33) | 10.4 | 48 | Tarn (81) | 5.9 | 50 |
| Hérault (34) | 10.8 | 32 | Tarn-et-Garonne (82) | 2.7 | 67 |
| Ille-et-Vilaine (35) | 8.4 | 74 | Var (83) | 13.6 | 54 |
| Indre (36) | 36.9 | 102 | Vaucluse (84) | 7.3 | 58 |
| Indre-et-Loire (37) | 15.3 | 60 | Vendée (85) | 6.4 | 83 |
| Isère (38) | 12.2 | 85 | Vienne (86) | 9.2 | 91 |
| Jura (39) | 23.1 | 92 | Haute-Vienne (87) | 6.7 | 204 |
| Landes (40) | 3.2 | 26 | Vosges (88) | 72.6 | 135 |
| Loir-et-Cher (41) | 19.6 | 70 | Yonne (89) | 28.1 | 68 |
| Loire (42) | 32.5 | 232 | Territoire de Belfort (90) | 51.1 | 137 |
| Haute-Loire (43) | 7.9 | 157 | Essonne (91) | 42.8 | 45 |
| Loire-Atlantique (44) | 12.2 | 65 | Hauts-de-Seine (92) | 69.3 | 34 |
| Loiret (45) | 15.3 | 55 | Seine-St-Denis (93) | 63.0 | 34 |
| Lot (46) | 12.7 | 88 | Val-de-Marne (94) | 88.7 | 46 |
| Lot-et-Garonne (47) | 3.6 | 69 | Val-D'Oise (95) | 59.5 | 41 |
| **Date** | **01/09/2020** | **P-value** | **0.0054** | **SPEARMAN CORREL.** | **−0.281** |

*Source*: data.gouv.fr

$\binom{n}{2}$. The Kendall correlation coefficient called Tau ($\tau$) is then calculated with the following formula:

$$\tau = \frac{S}{\frac{n(n-1)}{2}} = \frac{2S}{n(n-1)} \tag{1}$$

**Table 4**. Spearman correlation coefficients between average radon concentrations and mortality rates in France for 4 different dates in 2020.

| France | | | | | |
|---|---|---|---|---|---|
| Date (dd/mm/yyyy) | 11/06/2020 | 17/08/2020 | 01/09/2020 | 30/10/2020 | 12/12/2020 |
| Spearman Correlation Coef. | −0.278 | −0.280 | −0.281 | −0.263 | −0.137 |
| P-value | 0.006 | 0.006 | 0.006 | 0.010 | 0.184 |

**Table 5**. Screening indoor radon data from the EPA/State Residential Radon Survey of New Mexico conducted during 1988-89.

| County | *Meas. total | MEAN | **GEOM. MEAN | MEDIAN | SD | MAX | %>4 pCi/L | %>20 pCi/L |
|---|---|---|---|---|---|---|---|---|
| BERNALILLO | 406 | 3.7 | 2.7 | 2.6 | 3.5 | 27.0 | 28 | 1 |
| CATRON | 16 | 1.4 | 1.0 | 1.0 | 1.2 | 4.2 | 6 | 0 |
| CHAVES | 52 | 2.7 | 2.2 | 2.3 | 1.7 | 6.6 | 17 | 0 |
| CIBOLA | 6 | 2.3 | 1.8 | 2.3 | 1.5 | 4.7 | 17 | 0 |
| COLFAX | 91 | 6.0 | 3.8 | 3.9 | 11.5 | 105.4 | 49 | 3 |
| CURRY | 47 | 2.6 | 1.9 | 2.1 | 2.1 | 11.3 | 13 | 0 |
| DEBACA | 12 | 1.3 | 1.1 | 1.0 | 1.0 | 4.2 | 8 | 0 |
| DONA ANA | 86 | 1.8 | 1.4 | 1.3 | 1.4 | 9.0 | 7 | 0 |
| EDDY | 51 | 2.0 | 1.2 | 1.3 | 1.9 | 7.5 | 16 | 0 |
| GRANT | 60 | 2.1 | 1.3 | 1.5 | 2.1 | 13.4 | 10 | 0 |
| GUADALUPE | 8 | 1.3 | 1.0 | 1.1 | 0.8 | 2.7 | 0 | 0 |
| HARDING | 12 | 1.9 | 1.2 | 1.1 | 1.9 | 6.9 | 8 | 0 |
| HIDALGO | 18 | 3.7 | 2.8 | 3.4 | 2.8 | 12.5 | 39 | 0 |
| LEA | 50 | 1.6 | 1.1 | 1.1 | 1.4 | 7.6 | 6 | 0 |
| LINCOLN | 18 | 2.6 | 1.9 | 1.7 | 2.5 | 10.1 | 11 | 0 |
| LOS ALAMOS | 42 | 3.0 | 2.4 | 2.7 | 2.2 | 13.0 | 24 | 0 |
| LUNA | 49 | 3.8 | 2.5 | 2.4 | 4.6 | 27.7 | 22 | 2 |
| MCKINLEY | 53 | 6.0 | 2.8 | 3.2 | 13.0 | 87.3 | 34 | 6 |
| MORA | 17 | 4.6 | 3.5 | 3.9 | 3.2 | 11.5 | 41 | 0 |
| OTERO | 46 | 2.7 | 1.6 | 1.9 | 3.4 | 21.6 | 17 | 2 |
| QUAY | 10 | 3.2 | 2.7 | 2.6 | 1.8 | 6.0 | 30 | 0 |
| RIO ARRIBA | 72 | 3.4 | 2.3 | 2.2 | 4.0 | 24.7 | 21 | 1 |
| ROOSEVELT | 44 | 2.2 | 1.7 | 1.7 | 1.7 | 7.4 | 11 | 0 |
| SAN JUAN | 196 | 2.4 | 2.0 | 1.9 | 2.2 | 24.8 | 11 | 1 |
| SAN MIGUEL | 78 | 4.9 | 3.1 | 3.2 | 5.9 | 36.2 | 45 | 4 |
| SANDOVAL | 76 | 4.6 | 2.3 | 2.0 | 10.2 | 76.7 | 20 | 3 |
| SANTA FE | 73 | 4.6 | 3.2 | 3.5 | 3.8 | 21.6 | 41 | 1 |
| SIERRA | 41 | 1.3 | 1.0 | 1.0 | 0.9 | 3.9 | 0 | 0 |
| SOCORRO | 41 | 2.5 | 1.9 | 2.0 | 1.7 | 7.2 | 17 | 0 |
| TAOS | 47 | 6.3 | 3.8 | 4.7 | 6.6 | 31.4 | 57 | 4 |
| TORRANCE | 10 | 3.9 | 2.4 | 2.8 | 3.6 | 9.4 | 50 | 0 |
| UNION | 32 | 3.4 | 2.5 | 2.1 | 3.1 | 15.1 | 31 | 0 |
| VALENCIA | 25 | 1.9 | 1.8 | 1.7 | 0.8 | 3.6 | 0 | 0 |

Data represent 2-7 days of charcoal canister measurements from the lowest level of homes tested. *Total number of measurement. **Geometric means
Source: USGS-EPA Open-File Report 93-292-F-p.104.

$\tau$ varies between −1 and 1, it can be interpreted as a Spearman correlation coefficient, i.e. the closer it is to 1 (respectively −1), the more certain we are that there is a positive correlation, i.e. a variation in the same direction between the two variables (respectively negative). If $\tau$ is close to 0, the probability that there is no monotonic relationship between the two variables [mortality rate-indoor radon] is high.

In the case where there are ties, a variant of the Kendall correlation called $\tau$-b ($\tau b$) is used, whose formulations are somewhat more complex [23]. From the map of the percentages of excess of the 4 pCi/L threshold of radon, we created a table allowing to assign to each state of the USA, one of the five predefined classes, and put opposite the mortality rates [21], classified in alphabetical order (from Alabama to Wyoming). A sorting on the Mortality Rate column is

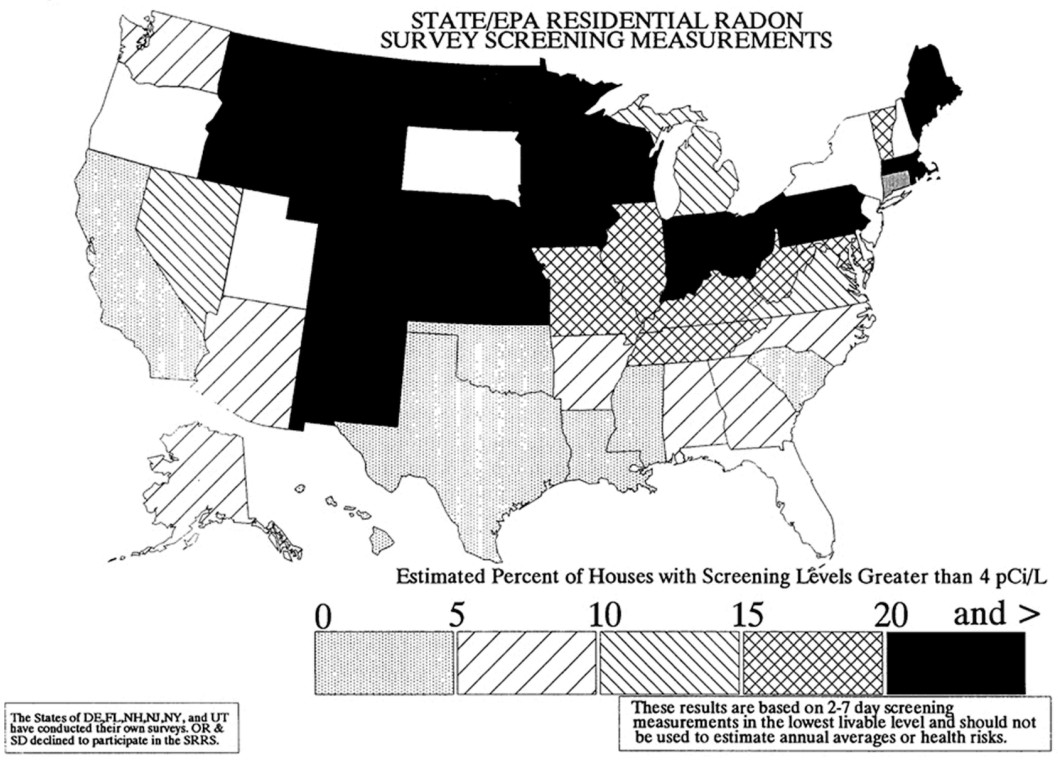

**Fig 6**. **USA - Percentage of homes exposed to radon levels above 4 pCi/L.** Source: U.S. Geological Survey and U.S. Environmental Protection Agency Open-File Report 93-292 (Public Domain).

performed, the states are then ranked according to an increasing rate. The Kendall correlation coefficient is calculated between the two columns Mortality Rate and Radon Map (Table 6) Classes. This correlation is tested for different arbitrary dates among the available data (01 July 2020, 19 August 2020, 19 September 2020, 30 November 2020). The following data analysis was generated using documentation from "Real Statistics Resources Pack software" Copyright (2013-2021), Zaiontz C [24]. The "Kendall's Correlation Testing with Ties" formulas provided for Excel were implemented and tested by us in Libre Office V7.1.3.2 (x64).

The Kendall correlation calculations were performed (Table 8) for the 4 same dates: 01/07/2020 at the end of the first wave in the USA, 19/08/2020, 19/09/2020 during the plateau that followed, and 30/11/2020 during the rise of the second wave and before the start of vaccinations.

For sufficiently large samples (n > 10), Tau follows a normal distribution, allowing for statistical evaluation of the results. Examination of the data in Table 8 revealed that the probabilities (p-values) were sufficiently low (<0.05) after 19/08/20 to conclude that the correlations presented for the U.S. are statistically significant. It is also noted that as the epidemic waves increase, the correlation weakens, as explained previously for France.

Thus, there is strong convergence of observations on independent data, suggesting that year-round exposure to indoor radon and its progeny results in lower mortality from COVID-19.

**Table 6**. Spearman correlation detailed data and calculation for 19/09/20 (excluding New York and New Jersey).

| USA States | State Code | *MR | **Radon map classes | USA States | State Code | *MR | **Radon map classes |
|---|---|---|---|---|---|---|---|
| Alaska | AK | 5.74 | 2 | California | CA | 38.02 | 1 |
| Vermont | VT | 9.30 | 4 | Arkansas | AR | 39.13 | 2 |
| Maine | ME | 10.34 | 5 | Ohio | OH | 39.46 | 5 |
| Wyoming | WY | 12.27 | 5 | Iowa | IA | 40.09 | 5 |
| Oregon | OR | 12.49 | 3 | New Mexico | NM | 40.39 | 5 |
| Utah | UT | 13.72 | 4 | Nevada | NV | 49.61 | 3 |
| Montana | MT | 14.60 | 5 | Alabama | AL | 49.70 | 2 |
| West Virginia | WV | 17.19 | 5 | Indiana | IN | 52.03 | 5 |
| Kansas | KS | 20.46 | 5 | Texas | TX | 52.16 | 1 |
| Wisconsin | WI | 21.45 | 5 | Georgia | GA | 60.75 | 2 |
| South Dakota | SD | 22.61 | 5 | Florida | FL | 61.86 | 1 |
| Nebraska | NE | 23.62 | 5 | South Carolina | SC | 61.92 | 1 |
| Oklahoma | OK | 23.83 | 1 | Pennsylvania | PA | 62.71 | 5 |
| Idaho | ID | 24.68 | 5 | Delaware | DE | 63.77 | 3 |
| Kentucky | KY | 25.45 | 4 | Maryland | MD | 64.11 | 4 |
| North Dakota | ND | 25.59 | 5 | Illinois | IL | 68.56 | 4 |
| Washington | WA | 27.97 | 2 | Michigan | MI | 69.79 | 2 |
| Missouri | MO | 31.04 | 4 | Arizona | AZ | 75.16 | 2 |
| North Carolina | NC | 31.08 | 2 | Washington D.C | DC | 87.71 | 2 |
| Tennessee | TN | 32.11 | 4 | Mississippi | MS | 94.38 | 1 |
| New Hampshire | NH | 32.21 | 5 | Rhode Island | RI | 102.70 | 1 |
| Virginia | VA | 35.03 | 3 | Louisiana | LA | 114.87 | 1 |
| Colorado | CO | 35.11 | 5 | Connecticut | CT | 125.99 | 1 |
| Minnesota | MN | 35.73 | 5 | Massachusetts | MA | 134.86 | 5 |
| **Date** | **19/09/2020** | | **SPEARMAN CORRELATION** | | | **-0.440** | |

*Deaths for 100.000 inhabitants; **USGS-EPA indoor Radon map classes (Fig 8). Source: github-nytimes and USGS-EPA Open-File Report 93-292-F.

---

**Table 7**. Spearman correlations for four successive randomly selected dates (excluding New York and New Jersey).

| | USA | | | |
|---|---|---|---|---|
| **Date (dd/mm/yyyy)** | 07/01/2020 | 19/08/2020 | 19/09/2020 | 30/11/2020 |
| **Spearman Correlation Coef.** | −0.165 | −0.406 | −0.440 | −0.258 |
| **P-value** | 0.261 | 0.004 | 0.002 | 0.076 |

Source: github-nytimes and USGS-EPA Open-File Report 93-292-F.

---

**Table 8**. Tau-b and p-value calculations for all U.S. states excluding New York and New Jersey for four selected dates.

Kendall Tau-b USA 48 États Radon Zones from Table 1: %>4pCi/L

| | 01/07/20 | 19/08/20 | 19/09/20 | 30/11/20 |
|---|---|---|---|---|
| n | 48 | 48 | 48 | 48 |
| C(n,2) | 1128 | 1128 | 1128 | 1128 |
| B | 486 | 582 | 588 | 526 |
| C | 349 | 267 | 260 | 328 |
| Tau-b | −0.141 | −0.322 | −0.335 | −0.202 |
| std error | 0.0997 | 0.0997 | 0.0997 | 0.0997 |
| z | −1.4132 | −3.22883 | −3.36057 | −2.02321 |
| Z-crit | 1.40507 | 2.80703 | 2.74778 | 1.95996 |
| P-value | 0.1576 | 0.0012 | 0.0008 | 0.0431 |

Source: Zaiontz C.(2020), Real Statistics Using Excel. www.real-statistics.com, implemented in LibreOffice 7.1.3.2 (x64).

## Discussion

### Study of High Background Natural Radioactive Areas (HBNRAs)

There are areas in the world where radon emissions are extremely high. These areas are identified under the name of "High Background Natural Radioactive Areas (HBNRAs)." They are regularly the subject of epidemiological studies concerning the frequency and nature of cancers in exposed populations, without any definitive conclusions being drawn as to the impact on the number of induced cancers [20]. It is interesting to study these areas in relation to COVID-19. Four well-identified areas include the Ramsar region in Iran on the Caspian Sea, the city of Guarapari in Brazil on the Atlantic coast, the city of Yangjiang in China on the South China Sea (southwest of Hong Kong), and the state of Kerala in India on the southwest coast of the Arabian Sea.

The natural radioactivity of the first three sites is rather localized (hot springs, beaches), making it difficult to draw conclusions regarding the pandemic. In particular, the extent of the use of radioactive beach sand for construction is not well known. However, the state of Kerala, which has a surface area of 38,852 km², a population of 35 million, and a density of more than 860 inhabitants per km² (compared to 324 in India and 413 in the Netherlands), shows interesting results. The first three COVID-19 cases in India appeared in Kerala in early February 2020, in three different districts, among students returning from Wuhan, China. The state government quickly implemented containment measures, but this did not prevent the disease from spreading rapidly throughout India.

The Health and Family Welfare Department of Kerala set up a daily bulletin with exemplary monitoring of the disease's spread (identification of deaths on a case-by-case basis, age, sex, date, origin of infection, and presence of comorbidities). As shown in Fig 7, the evolution of the number of deaths per week is very low and comparable to that of Finland (Figs 3 and 4).

Fig 8 shows the impressive results of the pandemic in Kerala, including confirmed cases, recoveries, and the low number of deaths. This suggests the potentially curative effect of radon by reducing deaths in severe cases, rather than preventing the number of severely ill patients.

### Statistical processing of French data

The observed correlation between radon exposure and COVID-19 mortality in France gradually weakened as the pandemic progressed, becoming negligible (correlation coefficient: –0.137) by December 2020. This could be explained by the fact that the population exposed to indoor radon in each territory remains relatively stable over these periods (i.e.,

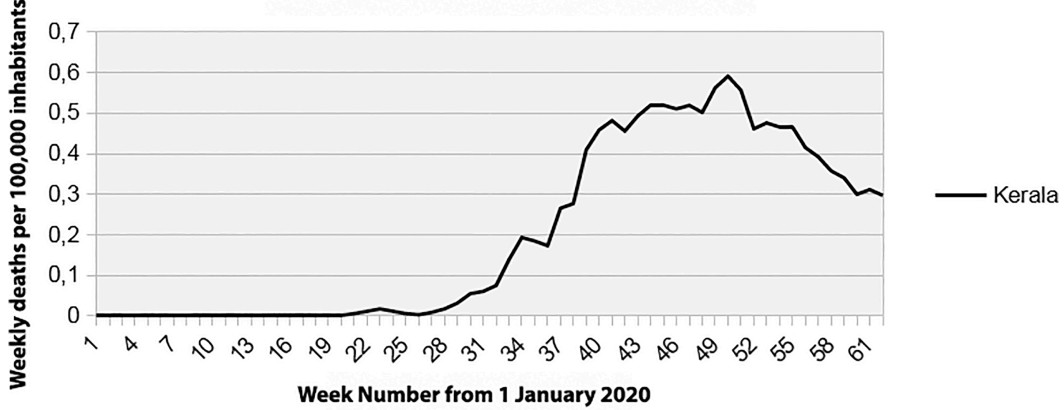

**Fig 7**. **Mortality rate for the State of Kerala in India, 2020.** Source: from the official Kerala COVID-19 dashboard (https://dashboard.kerala.gov.in/), adapted by the authors.

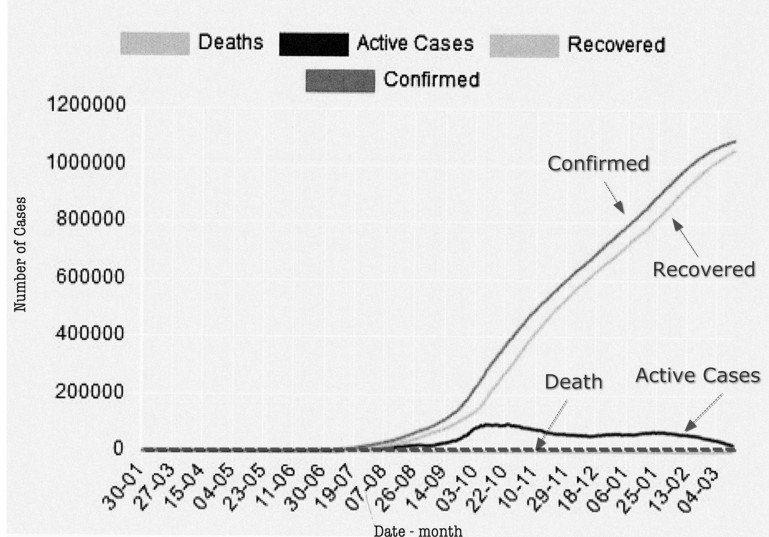

**Fig 8. Evolution of the pandemic in the state of Kerala, India, 2020).** Source: https://dashboard.kerala.gov.in/, adapted by the authors.

the number of people living in homes exposed to radon) while the number of contamination cases—and consequently deaths—continues to increase. This observation applies to other countries as well.

In observational epidemiological studies, matching methods are extensively used to estimate causal effects between exposures and outcomes to control for potential confounders and minimize bias [25]. The development of these methods is based on the causal effect concept, which is mathematically defined as the difference in the outcome under exposure for the same individual or unit ($\Delta Y_i = Y_i(1) - Y_i(0)$). These methods highlight areas where the covariate distribution lacks sufficient overlap between treatment and control groups, thus preventing over-reliance on extrapolation for exposure effect estimates [26]. Rubin and Thomas [27] demonstrated that analytic approximations for bias reduction in a linear combination of covariates, including outcomes, remain effective even when the assumption of normal distribution is violated. This formulation applies to aggregate data, such as schools or communities, and in this study, to regions.

We noted that regions in the USA and France with higher indoor radon levels often had lower population densities due to geography. To investigate this further, we carried out an analysis on French data, comparing 2 by 2 departments whose population densities are close to within 1%.

Two departments dpt1, dpt2 with respective densities de1 and de2, form a Ci pair if:

$$\Delta de_i = |de_1 - de_2| < 0.01 \max(de_1, de_2) \tag{2}$$

We thus obtained 38 Ci pairs among the 95 French départements. We observe that the members of these pairs are localized far from each others in France so that population moves are unlikely. For each of these pairs, we calculated the difference in indoor radon levels ra1 and ra2, the difference in mortality rates dc1 and dc2, and the difference in hospitalization rates ho1 and ho2. We multiplied these differences to calculate the signs of them ( $score_{dci}$ and $score_{hoi}$):

$$\Delta ra_i = (ra_1 - ra_2)_i, \quad \Delta dc_i = (dc_1 - dc_2)_i, \quad \Delta ho_i = (ho_1 - ho_2)_i \tag{3}$$

$$score_{dci} = \text{Signe}\left[\Delta ra_i \cdot \Delta dc_i\right], \quad score_{hoi} = \text{Signe}\left[\Delta ra_i \cdot \Delta ho_i\right] \tag{4}$$

If the score is negative, this means that, for the pair considered, the higher the indoor radon level, the lower the mortality or hospitalization rates.

The results are provided in S1 and S2 Tables in the supporting information section. For mortality rates, there are 21 pairs with negative scores and 17 with positive scores. For hospitalization rates, there are 23 pairs with negative scores and 15 with positive scores, indicating a stronger trend for hospitalization rates compared to mortality rates. These findings suggest a consistent effect of indoor radon exposure on health outcomes (mortality and hospitalization rates) among départements with very similar population densities. This observation more pronounced for hospitalization rates (referring to individuals leaving their homes) than for mortality rates (pertaining to individuals already hospitalized).

### Radon activity

Since ancient times, certain geological sites have been identified for their curative properties for various diseases. Many of these sites are still active today as spas, attracting tens of thousands of people per year worldwide and constituting a significant field for medical and wellness treatments. Many resorts now claim, in brochures explaining their therapies, medically controlled exposure to radon in several forms (inhalation, baths, etc.) as a health benefit for treating inflammatory rheumatoid arthritis.

Despite ongoing discussions among experts, it is recognized through the work of Franke A. [5] that radon exposure-based spa therapy is covered by different health systems in several European countries (Germany, Austria) as noted by Kabat G. [28]. A study on the concentrations of radon to which patients are exposed—just a few hours per day, over a few weeks, for one or more consecutive years—provides useful insights. One can examine the data available from thermal spas, where radon emanations, whether claimed or not, are almost always present. Améon R. [29] conducted a detailed study on this subject, as shown in Table 9.

It can be seen that in Bad Münster, Germany (a spa covered by the German health system), the concentration in the inhalation room is 128 kBq/m³. In the thermal galleries of Bad Gastein, Austria (a spa covered by the Austrian health system), the concentration is 166 kBq/m³. In Bagnères de Luchon, France (where radon exposure is not claimed in the therapy), it is 14 kBq/m³ in the spa caves.

**Table 9**. **Radon activity by volume (Av) in the air, measured in various spa rooms.**

| Rooms (Locaux) | $A_v$ (Bq/m³) | Note | Spa town | Reference |
|---|---|---|---|---|
| Administrative building | <40-410 | | Spain | Soto et Gomez (1999) |
| | 30-220 | | Bad Gastein (Austria) | Lettner et al. (1996) |
| Technical tunnel | >100 000 | | | Schmitz et Frische (1993) |
| Fountains hall | 280 | Relaxation room | Bad Münster (Germany) | Sansoni (1998) |
| | 1491 | | Misasa (Japan) | Morinaga et al. (1984) |
| Pools hall | 4 300-7 000 | Day-night | Rudas (Hungary) | Szerbin (1996) |
| | 22000 | | Bagnères-de-Luchon (France) | Grandpierre et al. (1962) |
| Aero baths | 1300 | Without ventilation | Evaux-les-Bains (France) | Améon et al. (2000) |
| | 150 | With ventilation | | Améon et al. (2001) |
| | 40-5 200 | Different locations | Spain | Soto et Gómez (1999) |
| | 40-5 000 | Outside-with treatments | Radenci (Yugoslavia) | Kobal et Renier (1987) |
| Spa tunnels | 16 600 | | Bad Gastein (Austria) | Uzunov et al. (1981) |
| | 2700 | | Misasa (Japan) | Morinaga et al. (1984) |
| Spa caves | 20000 | Summer value | Hospital (Hungary) | Szerbin (1996) |
| | 14000 | Summer value | Bagnères-de-Luchon (France) | Grandpierre et al. (1962) |
| Steam room | 7500 | | Evaux-les-Bains (France) | Améon et al. (2000) |
| | 103000 | Inhalation masks (radon therapy) | Las Caldas (Spain) | Soto et al. (1995) |
| Inhalation room | 12 8000 | | Bad Münster (Germany) | Sansoni (1998) |

Source: Ameon (2003); table format by authors.

What are the mechanisms behind radon's effects? The first possible explanation is the anti-inflammatory effect of low-dose ionizing radiation. In this regard, several recent studies have followed up on research conducted as far back as the 1940s, which explored the use of low-dose X-ray irradiation of the lungs to treat pneumonia [30–34]. Recent publications have reported success rates ranging from 10% to 30%. However, with the advent of antibiotics, which are much more effective and easier to use, this line of research was abandoned. Despite ongoing debate, these studies highlight the potential of nuclear radiation to mitigate the effects of COVID-19. One study by Clayton B. Hess [35] from Emory University Hospital, Atlanta, reported that in a cohort of 10 elderly hospitalized COVID-19 patients who were oxygen-dependent, seven patients recovered to room air in a median of 3 days, and were discharged in a median of 12 days. An article by Mortazavi [7] provides a comprehensive review of the results of low-dose radiation therapy (LD-RT) on COVID-19 patients. Taken together, the anti-inflammatory effects of low-dose radiation are comparable to those of steroid hormones given to many COVID-19 patients, but without the severe immunosuppressive effects seen with the latter [36].

Another possible mechanism for radon's effect could be the destruction of virions by alpha particles emitted during the decay of $^{222}$Rn. To understand this, it's essential to compare the number of SARS-CoV-2 virions present in the body of an infected individual with the number of $^{222}$Rn atoms inhaled in their environment. According to Madas et al. [37] and Sender et al. [38], there are approximately $10^7$ SARS-CoV-2 virions in the lungs of an infected person at any given time. If we consider the radon concentration in some departments in France, where it is relatively high (see Table 3), such as in Lozère, where it is 264 Bq/m³, we can calculate the number of $^{222}$Rn atoms. Using the formula N = ($T_{1/2}$ × A) / 0.693, where N represents the number of atoms, $T_{1/2}$ is $^{222}$Rn physical half-life, and A is the radioactivity, we arrive at a value of $1.25 × 10^8$ $^{222}$Rn atoms. Since an average adult inhales approximately 23 m³ of air per day [11], the total number of $^{222}$Rn atoms inhaled would be $2.9 × 10^9$, which is almost 300 times greater than the $10^7$ SARS-CoV-2 virions. This suggests that the probability of a virion being hit by an alpha particle emitted during $^{222}$Rn decay becomes significant. This interaction might provide an additional explanation for the lower COVID-19 mortality observed in regions with relatively high $^{222}$Rn concentrations. In addition to directly impacting the virions, alpha particles could also destroy infected cells in the lungs and airways, further contributing to the overall effect of $^{222}$Rn in reducing COVID-19 mortality. Numerous studies [37,39] discuss the localization of aerosol and droplet deposits, which may carry virions, and the distribution of $^{222}$Rn atoms in the respiratory tract, both attached and unattached. While alpha particles can damage healthy lung cells, the long-term consequences of this damage, such as the potential development of lung cancer, typically require decades to manifest and should be considered separately from the acute condition of COVID-19.

Finally, historical precedents for the use of ionizing radiation to treat infections have been reviewed [40]. More recently, molecularly targeted radiation, in the form of radioimmunotherapy, has been demonstrated to be effective in treating experimental fungal, bacterial, and viral infections [41]. Labeling antibodies that specifically bind to SARS-CoV-2 virions with alpha-emitting radionuclides, such as Astatine ($^{211}$At) or Lead ($^{212}$Pb), could be an effective way to deliver cytotoxic alpha particles to the surface of virions for their elimination.

## Limitations of the study

Indeed, the limitations of ecological studies are well known (Canadian Nuclear Safety Commission-Basic epidemiology concepts) but it is just a misguided scientific attitude to reject them regarding a pandemic which is not under control. John Snow, who is considered as the father of epidemiology, observed that the area served by water from the Broad Street fountain in London matched with the cholera pandemic (august 1854) found a correlation which led to much less fatalities. It will be a mistake to ignore the Hill's Criteria as causation analysis of this work (Hill's criteria are cited since 1965 with an exponential number of publications since 2000 (Google Scholar)). Going through these criteria, we find the following estimation of concordance as follow:

**Strength.** There are several correlations, moderately weak but repetitive (<-0,5), at countries population level between the binary data - "deaths rates" of the COVID-19 - and the exposures to an environmental data "the indoor Radon in

dwellings" which is an extremely complex variable of geogenic and anthropogenic origins (Fig 1). The difficulty lies finding geographical areas with indoor Radon dwellings measurements matching administrative areas with daily recorded COVID-19 deaths rates. The analysis covers 10 countries having such property over 3 continents with populations close to half billion individuals during the first year of the pandemic. The repetitiveness shows a possible association.

**Consistency.** The fact that both the longitudinal observed values and the correlations values at different dates are repeatable highlights a possible association. All the data concerns the spread of the pandemic which started everywhere nearly at the same time in March 2020 (first reported cases) up to mid-December 2020 when massive vaccination started.

**Specificity.** The existence of a short-term incidence (within month) between the indoor Radon exposure and the deaths rates has no common link with the long term (tenths of years) lung cancerogenesis of it.

**Temporality.** Indoor Radon dwellings is a lifelong type of exposure which was already affecting the populations in the concerned areas when the pandemic started.

**Biological gradient.** Both, Table 2 and the correlations show that an increase in the indoor Radon dwelling exposure show a monotonic negative relationship with deaths rates.

**Plausibility and coherence.** The effect of radiations to enhance the immunitary system and killing virus and microbes is widely recognized in the literature.

**Analogy.** Thermal waters have been used for centuries to improve certain diseases, particularly those of genetic origin; X-rays empiric treatment of pneumonia with 30% of success before the discovery of the penicillin; Clinical tests for the treatment of the COVID-19 using X-Rays with spectacular fast recoveries in several countries.

Since the first work of Hill's in 1965, several authors have criticized Hill's criteria on the basis that several epidemiological studies gave contested conclusions, some of these are reported, following the work of Rothman [42] and summarized in the work of George Davey Smith [43].Regarding environmental epidemiology, where many parameters are not known and not measured, the use of randomized control trials (RCT) can not be used, and other approaches must be used as exposed in Pearce and al. [44]. One of his approaches, in our context, might be the triangulation of epidemiological evidence (as quoted in the Analogy, Plausibility and Coherence criteria). "Criteria for its use in causal inference in epidemiology have been proposed recently, and these specify that results from at least two (but ideally more) methods that have differing key sources of unrelated bias be compared. If evidence from such different epidemiologic approaches all point to the same conclusion, this strengthens confidence that is the correct causal conclusion, particularly when the key sources of bias of some of the approaches would predict that the findings would point in opposite directions" [44].

## Data reliability

Except for France and Kerala where the data come from the governmental agencies, all other data come from the Johns Hopkins University (JHU). This university has established the Corona Resources Center (CRC) with the support of the Blomberg Philanthropies and the Stavros Niarcos Foundation (SNF) to collect and analyze the best available data on cases, deaths, tests, hospitalizations and vaccines to assist all stakeholders in the pandemic. In 2021, Research America presented CRC with the "Meeting the Moment for Public Health" award. Data is collected daily from every country in the world. Mortality results are provided in terms of the number of deaths per 100,000 population (country population). It is the latter ratio that we use since it is consistent with the radon exposure data that concern the whole population in each territory. It is clear that data from countries with high quality health services and statistics are as reliable as possible. This is especially true of the countries we have selected for this study. It is quite remarkable that the data for Kerala in India are also considered extremely reliable by other authors and have been used to reconsider the global statistics of deaths provided by India in general [45]. The CRC has reservations about the accuracy of some of the data provided by some countries.

**Table 10. Spearman correlation results for the same dates as view previously.**

| Spearman Correlation: Radon and Hospitalized Persons Average Rank | | | | | |
|---|---|---|---|---|---|
| Date (dd/mm/yyyy) | 11/06/2020 | 17/08/2020 | 01/09/2020 | 30/10/2020 | 12/12/2020 |
| Spearman Correlation Coef. | −0.313 | −0.327 | −0.333 | −0.361 | −0.228 |

## Other environmental factors and possible bias

The relationship between the disease contracted and its fatal outcome is influenced by various biases, many of which are difficult to analyze, such as contamination, age and comorbidities. Contamination, roughly estimated by population density, has already been discussed in the matched analysis.

With regard to age, it is particularly relevant to examine the impact of radon exposure on the elderly. Research consistently shows that COVID-19 patients are more likely to suffer serious illness than younger individuals, often resulting in hospitalization. Therefore, examining mortality rates alone may be insufficient to attempt to estimate this bias. It may be more instructive to compare hospitalization rates rather than mortality rates. French daily data, which record both discharges home and deaths in hospital, can be used to analyse hospitalization rates.

We compared indoor radon data from different departements with the percentage of people hospitalized. Table 10 shows a similar trend for the hospitalization rates as for the mortality rates during the period studied, with a correlation coefficient close to −0.32. Notably in France, the correlation between the departements indoor radon exposure and the proportion of people aged 65 and over living there is 0.595. This indicates that a higher percentage of older adults are present in departments with higher indoor radon concentrations due to geographical and sociological contexts (corroborating results also observed in the USA).

Analyzing density presents more complexity than addressing age bias. The mean density within a department may not serve as a reliable indicator of contamination levels. However, it remains worth exploring these potential biases implications. Future investigations should incorporate comprehensive data on a range of potential confounders to strengthen the results. Although this study provides valuable information, it is important to recognize its limitations. Therefore, we recommend further research to explore this topic in greater depth.

## Conclusion

Given the complexity of the various parameters that may influence the results of this research, along with its potentially disruptive nature, it is essential to exercise caution when drawing causal inferences about Radon exposure and its potential therapeutic effects on COVID-19 outcomes. Nevertheless, our study has shown a positive correlation between residential exposure to Radon and reduced morbidity from COVID-19 during the pre-vaccination period across three continents. This finding aligns with both historical and recent clinical observations indicating that low-dose ionizing radiation can reduce lung inflammation caused by infectious agents. However, the molecular mechanisms underlying this phenomenon require further investigation. Additionally, these results suggest an intriguing possibility for the future use of molecularly targeted radiation, such as radio-immunotherapy, in the treatment of COVID-19.

## Supporting information

**S1 Table. Analysis of Mortality rate and Radon Exposure in Relation to Population Density.** This table provides a detailed breakdown of the statistical analysis, exploring the relationship between population density, radon exposure, and mortality rates across various regions in France using a matching pair differential.
(PDF)

**S2 Table. Analysis of Mortality rate and Radon Exposure in Relation to Hospitalization Number in France.** This table provides a detailed breakdown of the statistical analysis, exploring the relationship between population density, radon exposure, and the total hospitalization number across various regions in France using matching pair differential. (PDF)

## Author contributions

**Conceptualization:** Jean-François Coudert, Ekaterina Dadachova, Gilles Maignant.

**Data curation:** Jean-François Coudert, Stephanie Jonathan.

**Formal analysis:** Jean-François Coudert, Gilles Maignant, Stephanie Jonathan.

**Methodology:** Jean-François Coudert, Gilles Maignant.

**Validation:** Jean-François Coudert, Ekaterina Dadachova, Gilles Maignant, Stephanie Jonathan.

**Visualization:** Jean-François Coudert, Stephanie Jonathan.

**Writing – original draft:** Jean-François Coudert, Ekaterina Dadachova, Gilles Maignant.

**Writing – review & editing:** Jean-François Coudert, Ekaterina Dadachova, Gilles Maignant, Stephanie Jonathan.

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
