## [Decision Letter · Decision Letter 0]

27 May 2025

PONE-D-24-57007Radon exposure and COVID-19 mortality in pre-vaccination period: what links might exist?PLOS ONE

Dear Dr. Jonathan,

Thank you for submitting your manuscript to PLOS ONE. The reviewers recomended your mnauscript for publication in PLOS ONE subject to minor revision listed below.  Therefore, we invite you to submit a revised version of the manuscript that addresses the points raised during the review process.

 Thank you for submitting your manuscript to PLOS ONE. The reviewers recomended your mnauscript for publication in PLOS ONE subject to minor revision listed below.  Therefore, we invite you to submit a revised version of the manuscript that addresses the points raised during the review process.

We look forward to receiving your revised manuscript.

Kind regards,

Said Muhammad

Academic Editor

PLOS ONE

Journal Requirements:

2. We note that Figures 2 & 8 in your submission contain [map/satellite] images which may be copyrighted. All PLOS content is published under the Creative Commons Attribution License (CC BY 4.0), which means that the manuscript, images, and Supporting Information files will be freely available online, and any third party is permitted to access, download, copy, distribute, and use these materials in any way, even commercially, with proper attribution. For these reasons, we cannot publish previously copyrighted maps or satellite images created using proprietary data, such as Google software (Google Maps, Street View, and Earth). For more information, see our copyright guidelines: http://journals.plos.org/plosone/s/licenses-and-copyright.

a. You may seek permission from the original copyright holder of Figures 2 & 8 to publish the content specifically under the CC BY 4.0 license. 

Additional Editor Comments (if provided):

Manuscript Number: PONE-D-24-57007

Radon exposure and COVID-19 mortality in pre-vaccination period: what links might exist?

Dear Stephanie Jonathan, M.D,  

I have completed my evaluation of your manuscript. The reviewers recommend acceptance of your manuscript following minor revision and modification. I invite you to resubmit your manuscript after addressing the comments below. Please resubmit your revised manuscript within due time.

When revising your manuscript, please consider all issues mentioned in the reviewers' comments carefully: please outline every change made in response to their comments and provide suitable rebuttals for any comments not addressed. Please note that your revised submission may need to be re-reviewed.

Reviewers' comments:

Reviewer's Responses to Questions

**Comments to the Author**

1. Is the manuscript technically sound, and do the data support the conclusions?

Reviewer #1: Yes

Reviewer #2: Yes

2. Has the statistical analysis been performed appropriately and rigorously?

Reviewer #1: Yes

Reviewer #2: Yes

3. Have the authors made all data underlying the findings in their manuscript fully available?

Reviewer #1: Yes

Reviewer #2: Yes

4. Is the manuscript presented in an intelligible fashion and written in standard English?

Reviewer #1: Yes

Reviewer #2: Yes

5. Review Comments to the Author

Reviewer #1: This cross-sectional ecological study investigated the association between indoor radon exposure and COVID-19 mortality rates across eight countries, including several European nations, the United States, and the State of Kerala, India, during the pre-vaccination period. The results demonstrated a consistent negative correlation, indicating that regions with higher radon concentrations tended to experience lower COVID-19 mortality rates. Although the findings are not conclusive, the data suggest a potential mitigating effect of radon exposure on COVID-19 mortality. This unique research has the potential to pave the way for further studies, offering significant benefits to public health. Additionally, the manuscript is well-written, and the research is scientifically robust. Based on its merit and well-executed design, I recommend acceptance of this manuscript without any modifications.

Reviewer #2: The manuscipt is well written in clear English. However there are minor gramattical suggestions highlited in the reviewed manuscript.

Also the introduction section is very lengthy. some sections may be shifted to discussion section, like data and discussion of Kerala.

6. PLOS authors have the option to publish the peer review history of their article (what does this mean?). If published, this will include your full peer review and any attached files.

Reviewer #1: No

Reviewer #2: No

---

## [Author Response · Author response to Decision Letter 1]

27 Oct 2025

Response to Reviewers – Manuscript PONE-D-24-57007

Title: Radon exposure and COVID-19 mortality in pre-vaccination period: what links might exist?

Dear Academic Editor and Reviewers,

We sincerely thank you for your time, your positive evaluations, and the constructive comments provided during the review of our manuscript. We are pleased that both reviewers found the manuscript scientifically sound, well-written, and suitable for publication pending minor revisions.

Please find below our detailed point-by-point responses. All modifications have been incorporated into the revised manuscript and tracked accordingly in the submitted version with changes.

General Reviewer Feedback

Reviewer 1:

“This unique research has the potential to pave the way for further studies... Based on its merit and well-executed design, I recommend acceptance of this manuscript without any modifications.”

Response:

We thank Reviewer 1 for the positive feedback and support for our manuscript. No specific changes were requested.

Reviewer 2:

“The manuscript is well written in clear English. However, there are minor grammatical suggestions highlighted in the reviewed manuscript. Also, the introduction section is very lengthy. Some sections may be shifted to the discussion section, like data and discussion of Kerala.”

Response:

We appreciate Reviewer 2’s helpful suggestions. In response:

1. Grammar: We have carefully addressed all grammatical suggestions as indicated in the reviewer’s annotated file. (Line 85)

2. Introduction length: We agree the introduction was too long. We have shortened it by relocating the Kerala-specific background and interpretation to the Discussion section (lines 236–263) where it fits better with our results. Additionally, some parts related to the statistical study using French data have been moved from the Materials and Methods to the Discussion (lines 265–271). Figure numbering has been updated accordingly.

3. Clarification in the abstract:

o Original: “The findings revealed a consistent negative correlation between higher radon concentrations and lower COVID-19 mortality rates.”

o Revised: “The findings revealed a consistent negative correlation between radon concentrations and COVID-19 mortality rates, indicating that higher radon concentrations were associated with lower mortality rates.”

Editorial and Journal Requirements

1. Style & Formatting:

We have reviewed the PLOS ONE formatting guidelines and updated the manuscript and files to meet the style requirements, including proper file naming.

2. Figures 2 & 8 — Map image copyright:

Figure 2 (European Atlas of Natural Radiation):

We confirm that Figure 2 is adapted from the European Atlas of Natural Radiation by the European Commission’s Joint Research Centre (JRC). According to the European Commission Decision of 12 December 2011 on the reuse of Commission documents, this material is available under the Creative Commons Attribution 4.0 International (CC BY 4.0) licence unless otherwise indicated.

Figure 1. Description of the phenomenon involved in the risk evaluation of indoor radon. We received permission from the author, Peter Bossew, to reuse this figure, and we have attached the permission document accordingly.

Figure 8 (USGS–EPA Open-File Report 93-292 previously figure 8. After revisions, it is figure 6):

We confirm that Figure 8 is reproduced from the USGS–EPA Open-File Report 93-292, which is an official publication by U.S. federal agencies. Under U.S. law (17 U.S.C. § 105), works created by the U.S. government are in the public domain and are not subject to copyright restrictions.

To comply with PLOS ONE’s policy, we have:

o Verified that this material is in the public domain.

o Updated the figure caption to include a clear public domain notice and correct source attribution.

We hope this clarifies the licensing status of both figures and we remain available should the editorial team require any further information.

3. References:

We have systematically checked all 45 references in five batches. None are retracted or under expression of concern. All DOIs are valid and the articles remain indexed on PubMed, PMC, and publisher sites.

4. Minor reference updates:

We have updated a few references for completeness and corrected minor formatting issues. These changes are tracked in the revised manuscript:

o Page 5 – source: ECDC, 2020

o Line 157 – INSEE, 2020

o Figure 6 – X and Y variable are now labelled

We hope that our responses and revisions meet your expectations and would like to thank you again for your time and consideration.

Kind regards,

Stephanie Jonathan, M.D.

(On behalf of all co-authors)

---

## [Editor Report · Decision Letter 1]

6 Nov 2025

Radon exposure and COVID-19 mortality in pre-vaccination period: what links might exist?

PONE-D-24-57007R1

Dear Dr. Jonathan,

We’re pleased to inform you that your manuscript has been judged scientifically suitable for publication and will be formally accepted for publication once it meets all outstanding technical requirements.

Kind regards,

Said Muhammad

Academic Editor

PLOS ONE
---

## [Editor Report · Acceptance letter]

PONE-D-24-57007R1

PLOS ONE

Dear Dr. Jonathan,

I'm pleased to inform you that your manuscript has been deemed suitable for publication in PLOS ONE. Congratulations! Your manuscript is now being handed over to our production team.

Kind regards,

on behalf of

Dr. Said Muhammad

Academic Editor

PLOS ONE